# Successful Treatment of Vertebral Osteosarcoma in a Cat Using Marginal Surgical Excision and Chemotherapy

**DOI:** 10.3390/vetsci9070315

**Published:** 2022-06-23

**Authors:** Antonio Giuliano, Virginie De Busscher, Diane D. A. Lu, Karen W. L. Ng, Julia A. Beatty

**Affiliations:** 1Veterinary Medical Centre, City University of Hong Kong, Hong Kong, China; debusscherv@yahoo.com (V.D.B.); dianelu@cityuvmc.com.hk (D.D.A.L.); karen.ngvet@gmail.com (K.W.L.N.); 2Department of Veterinary Clinical Sciences, Jockey Club College of Veterinary Medicine, City University of Hong Kong, Hong Kong, China; julia.beatty@cityu.edu.hk; 3Centre for Animal Health and Welfare, City University of Hong Kong, Hong Kong, China

**Keywords:** cats, feline, axial osteosarcoma, toceranib phosphate

## Abstract

A three-year-old male neutered Norwegian Forest cat was referred for bilateral ambulatory paraparesis and spinal pain. On magnetic resonance imaging (MRI), a mass involving the right epaxial muscles with vertebral canal invasion and causing marked extradural spinal cord compression was identified. At surgery, the mass was debulked and a right hemilaminectomy was performed. Histopathology was diagnostic of fibroblastic osteosarcoma. Residual osteolytic lesions of the osteosarcoma were present at the level of the spinous process of the second lumbar vertebra. Four cycles of adjuvant doxorubicin chemotherapy were administered followed by oral toceranib phosphate. Neurological signs improved gradually over weeks to months and the lesion in the spinous process was no longer visible on radiographs. At one year from diagnosis, an MRI of the T3-L3 (3rd thoracic vertebra to the 3rd lumbar vertebra) spinal region and a whole-body computer tomography (CT) scan found no evidence of the osteosarcoma in the spine or of any metastasis. All medications were stopped and, at the time of writing 16 months later, the patient is neurologically normal with no signs of cancer recurrence. This is the first case report documenting the complete resolution of vertebral osteosarcoma lesions after treatment with doxorubicin followed by toceranib phosphate. The treatment also prevented tumor recurrence and was associated with an exceptionally long-term survival time.

## 1. Case Description

A three-year-old male neutered Norwegian Forest cat was referred for ambulatory paraparesis and spinal pain. The neurological signs had been slowly progressive over the previous 4–6 weeks.

On neurological examination, the abnormal findings were confined to the pelvic limbs. The cat sat with the right pelvic limb splaying sideways, and the hindquarters were moderately crouched during standing. An abnormal gait was characterized as an asymmetrical paraparesis worse in the right pelvic limb. Bilateral postural reaction deficits were identified; extensor postural thrust was absent in the right pelvic limb and reduced in the left, hopping was reduced in the right pelvic limb, and proprioception was absent in both pelvic limbs. Muscle tone and spinal reflexes were normal. Pain was evident on palpation of the lumbar spine, and a mild bulge was palpated in the right cranial lumbar area. The neurological findings were consistent with a lesion localized to T3-L3 (3rd thoracic vertebra to the 3rd lumbar vertebra) spinal cord segment. Physical examination was otherwise unremarkable.

## 2. Materials and Methods

Hematology, biochemistry and urinalysis were within normal limits, and serological testing for feline immunodeficiency virus, feline leukaemia virus, and Toxoplasma gondii (immunoglobulin G and immunoglobulin M) were all negative. On magnetic resonance imaging (MRI) of the T3-L3 spinal cord, a multilobulated, well-demarcated mass (3 cm × 1.5 cm × 1.25 cm) was seen in the right epaxial muscles invading the second and third lumbar vertebrae and causing extradural spinal cord compression at L2 (2nd lumbar) and L2/3 (2nd to 3rd lumbar) intervertebral space (Figure 1).

The relatively homogeneous mass was T1W, T2W, and T2W IDEAL (fat suppression sequence) hyperintense and presented strong diffuse contrast enhancement on MRI. The spinous process and dorsal lamina of the second lumbar vertebra had a thin, irregular cortex and contrast enhancement of the medulla, suggestive of bone invasion and lysis. An irregular spiculated periosteal reaction was seen at the left dorsal aspect of the lamina of L2 (Figure 2 and Figure 3).

The mass extended around the articular processes of L2/3 (right more than left) and in the right epaxial muscles up to the caudal aspect of L3 (3rd lumbar vertebra). Some lysis of the right articular processes of L2/L3 was suspected, and marked invasion of the vertebral canal was observed. (Figure 4).

The mass occupied up to two thirds of the canal at the dorsal and lateral aspects of the compressed spinal cord, from the mid aspect of L2 to the cranial aspect of L3. Expansion of the vertebral canal due to the presence of the mass just cranial to L2/3 could be visualized. Mild dilation of the central canal and subtle T2W spinal hyperintensity of the spinal cord suggestive of mild oedema was evident around the site of extradural compression. (Figure 5).

Presurgical thoracic radiographs, in three views, to screen for metastatic disease were unremarkable. At surgery, the mass was debulked, and a right L2–L3 hemilaminectomy was performed to decompress the spinal cord. The cat recovered well, and neurological signs improved in the following weeks, apart from mild spinal hyperesthesia at the surgical site and reluctance to jump. The patient was started on pain management with gabapentin 4 mg/kg twice a day and, 2 weeks post-surgery, prednisolone 0.5 mg/kg daily was added and progressively reduced to every other day. The hyperesthesia responded well to the treatment, but occasional discomfort was suspected by the owner during the following 5 months.

Histopathology of the mass revealed an unencapsulated, infiltrative, and poorly demarcated neoplasm with several irregular, anastomosing lamella bone spicules with lacunae of osteocytes. Neoplastic cells were spindle to polygonal with indistinct margins, had pale eosinophilic cytoplasm, fusiform nuclei, and two distinct nucleoli (Figure 6 and Figure 7). Neoplastic cells extended to all the surgical margins.

Adjuvant chemotherapy comprised intravenous doxorubicin, 1mg/kg every 3 weeks for four cycles in total. Due to the lytic lesions in the vertebral spinous process of L2, the patient was also started on alendronate 1 mg/kg twice a week. Gabapentin and prednisolone treatments were progressively reduced, then stopped 8 weeks post-surgery due to resolution of spinal pain. Due to the persistence of the L2 lytic lesions on radiography, the alendronate was continued. Every 3 weeks before doxorubicin administration, the cat was examined and hematology, biochemistry, and urinalysis were performed. Spinal radiographs were repeated at 6 weeks and at the end of the doxorubicin chemotherapy protocol, 12 weeks later.

On completion of doxorubicin treatment, toceranib phosphate 10 mg was prescribed once every Monday, Wednesday, and Friday. Patient monitoring was continued every 4 and 6 weeks with clinical examination, hematology, biochemistry, and urinalysis, including urine protein to creatinine ratio (UPC) and spinal radiographs. The lytic lesions in the L2 gradually improved with complete resolution one year after surgery.

## 3. Results

A whole-body CT scan and an MRI of the T3-L3 spine were repeated 12 months post-surgery. On both MRI and CT scans, the only significant abnormality was a focal bone defect involving the articular facets on the right side of L2–L3, which was attributed to the previous hemilaminectomy. Specifically, no evidence of the previously diagnosed osteosarcoma was identified, and the complete resolution of the osteolytic lesions in L2 was also confirmed (Figure 8). Medical treatment was stopped, and at the time of writing, 16 months after diagnosis, the cat is clinically well and neurologically normal with no signs of cancer recurrence.

## 4. Discussion

Primary bone tumors are rare in cats, but osteosarcoma is the most common tumor type, affecting both the appendicular and axial skeleton [1,2]. Osteosarcoma usually affects older cats with a reported median age of 8–10 years, although young cats can be also affected [3,4]. Whether the age of the patient in this report could have influenced the prognosis is not clear. In cats with osteosarcoma, there are no reported differences in prognosis between young and older patients [3,4]. In people, young patients with osteosarcoma have better outcomes compared to older people; however, this may be related to a more aggressive treatment reserved for younger patients [5,6]. In dogs with appendicular osteosarcoma, there is no clear difference in survival between young and old patients, but interestingly, young dogs could have a worse prognosis [7]. In canine axial osteosarcoma, age is not shown to significantly affect survival [8].

Appendicular osteosarcoma has a low metastatic rate in cats compared with dogs, and amputation is often curative, so adjuvant treatments are usually not required in feline cases [4]. Axial osteosarcomas are more difficult to resect completely, and prognosis is less favorable than for appendicular osteosarcoma [4]. The mean survival for vertebral osteosarcoma treated with surgery, with or without radiotherapy, is around 6 months, and local tumor progression is the principal cause of treatment failure [4,9].

Evidence of the effect of adjunctive, post-surgical radiotherapy or chemotherapy for feline axial osteosarcoma on outcomes is lacking. Only a few case reports and retrospective studies have evaluated the role of adjuvant treatment for axial and extra-skeletal osteosarcoma, but some survival benefits have been hypothesized [4,10,11]. Similarly, in dogs, the benefit of adjuvant treatment for spinal osteosarcoma has yet to be formally evaluated, but improvement in survival with a multimodal treatment approach is likely [8,12,13]. Radiotherapy was not available locally at the time this case was diagnosed. Thus, doxorubicin followed by toceranib was used to attempt to reduce tumor recurrence. Doxorubicin is reported to achieve clinical objective response and possibly to decrease recurrence rate and increase survival in feline soft tissue sarcomas and in particular in feline injection site sarcoma [14,15,16,17,18]. Its effect, if any, in feline osteosarcoma is not known. Toceranib, a multi-kinase inhibitor with anti-angiogenic property, is effective as a single agent in a wide range of solid tumors in dogs [19,20]. One of the cellular targets of toceranib is vascular endothelial growth factor receptors (VEGFR), and high VEGFR expression has been demonstrated in soft tissue sarcoma in dogs [21]. Targeting VEGFR could decrease blood supply to the tumor, potentially reducing tumor growth. Toceranib also inhibits KIT (stem cell factor) and PDGFR (platelet derived growth factor), which have direct cytotoxic effects and contribute to the anti-angiogenic effect [22]. However, Toceranib has not been proven to be effective in the local control of the disease in injection site sarcoma [22]. The efficacy of toceranib in more slowly growing sarcomas, such as feline osteosarcoma, is not yet known. The potential benefits of toceranib in feline osteosarcoma relate to its broad anti-angiogenic properties, especially in tumors with high VEGFR expression.

Bisphosphonates, such as alendronate, are a class of compounds used to treat osteoclast-mediated bone loss. These compounds are known to inhibit osteoclast activity, and also to exhibit anti-angiogenic properties and anticancer effects in vitro [23].

Bisphosphonates have multiple direct growth inhibitory effects on osteosarcoma cells through the activation of apotosis and inhibition of proliferation. Bisphosphonates have anti-angiogenic activity and direct inhibition on endothelial cells via inhibition of VEGF (vascular endothelial growth factor) and VEGFR [23]. However significant clinical tumor response or survival improvement in both dogs and cats with osteosarcoma have never been reported.

In this patient, alendronate in combination with doxorubicin followed by toceranib treatment may have contributed to the resolution of the osteosarcoma bone lesions.

In this case report, in contrast to previous reports [4,9], the tumor did not recur, and the cat remains clinically well 16 months from diagnosis. It is interesting to note that the neoplastic lytic lesions affecting L2 resolved completely after toceranib treatment and were no longer visible on either radiographs, and later MRI and CT scan, suggesting a complete response to toceranib. The efficacy of toceranib in axial osteosarcoma in cats has never been reported, and it is possible that this drug could be beneficial in unresectable osteosarcoma or, as in this case, in the treatment of residual disease after debulking surgery.

## 5. Conclusions

Incomplete resection or debulking surgery for feline axial osteosarcoma is generally considered a negative prognostic factor, and survival is usually limited due to local recurrence [3,4,9]. Here, adjunctive alendronate and doxorubicin chemotherapy, followed by toceranib, are likely to have contributed to the excellent outcome and the complete resolution of the residual tumor. However, a longer follow up is needed, as late recurrence of the osteosarcoma is still possible.

Evidence gained from prospective, controlled clinical studies would be valuable to guide treatment decision-making in the future.

## Figures and Tables

**Figure 1 vetsci-09-00315-f001:**
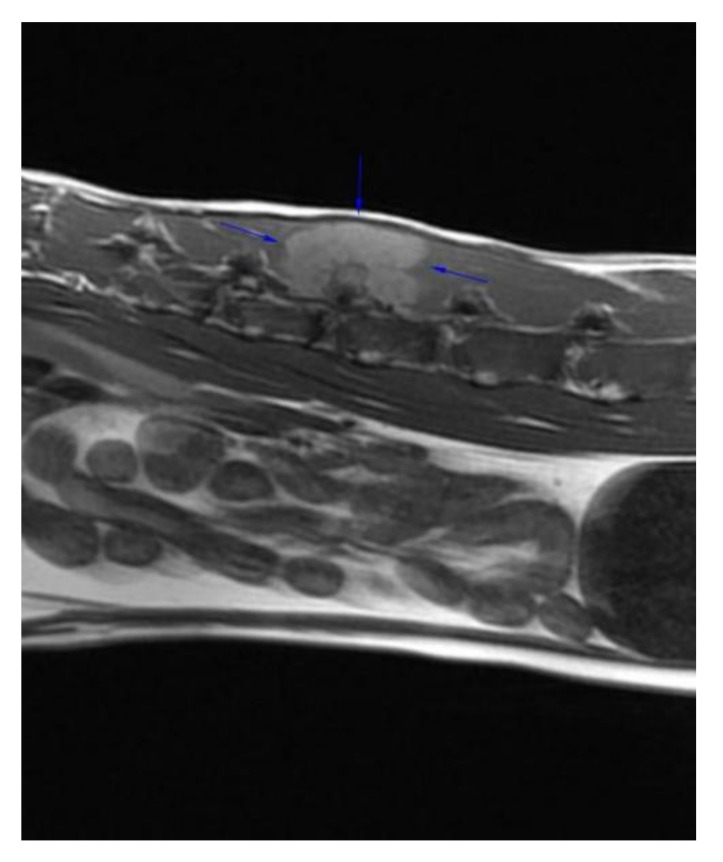
Right para-sagittal pre-contrast T1W image of the mass showing the T1W hyperintense mass in the right epaxial muscle dorsal to L2 and L3 (blue arrows).

**Figure 2 vetsci-09-00315-f002:**
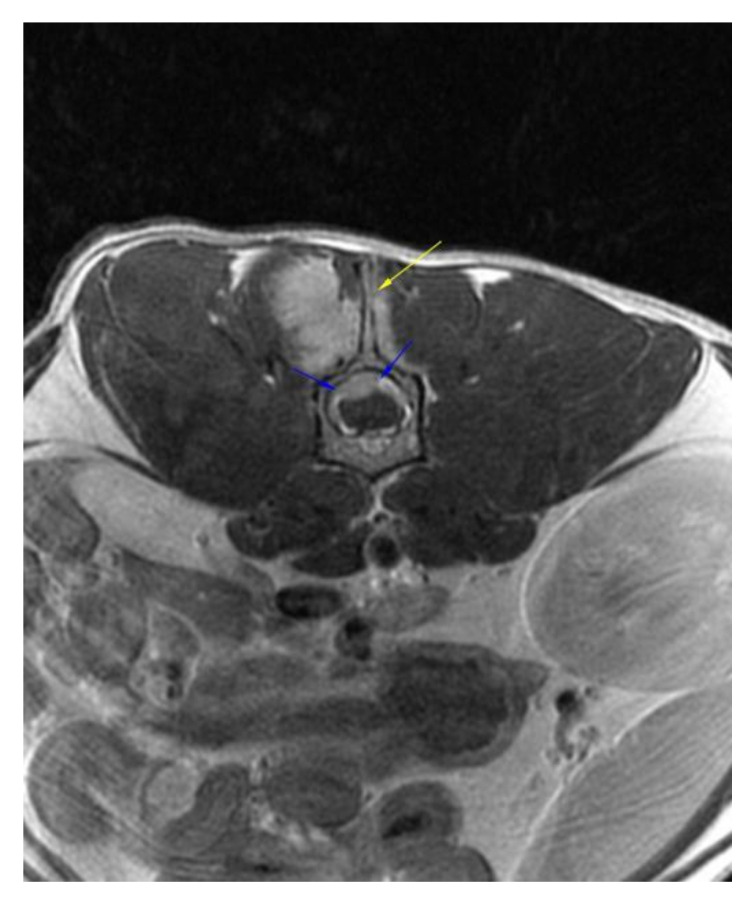
Post-contrast transverse T1W image showing contrast enhancement, thinning, and irregularity of the spinous process (yellow arrow) and dorsal lamina of L2 together with the invasion of the vertebral canal by the mass (blue arrows).

**Figure 3 vetsci-09-00315-f003:**
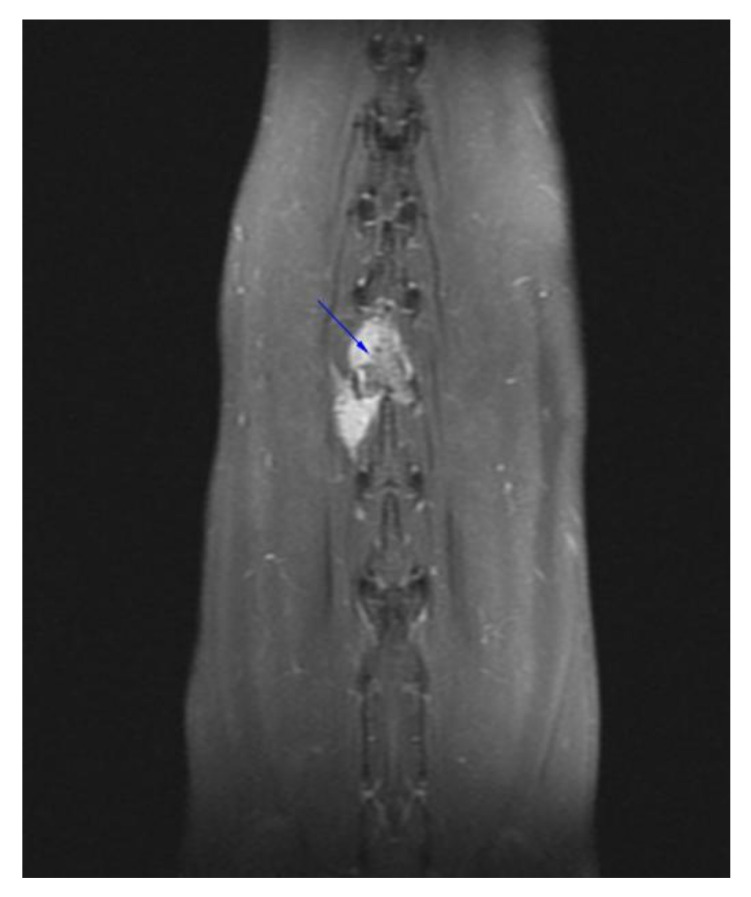
Post-contrast dorsal T1W image showing the contrast enhancement of the lamina of L2 in the middle of the mass (blue arrow).

**Figure 4 vetsci-09-00315-f004:**
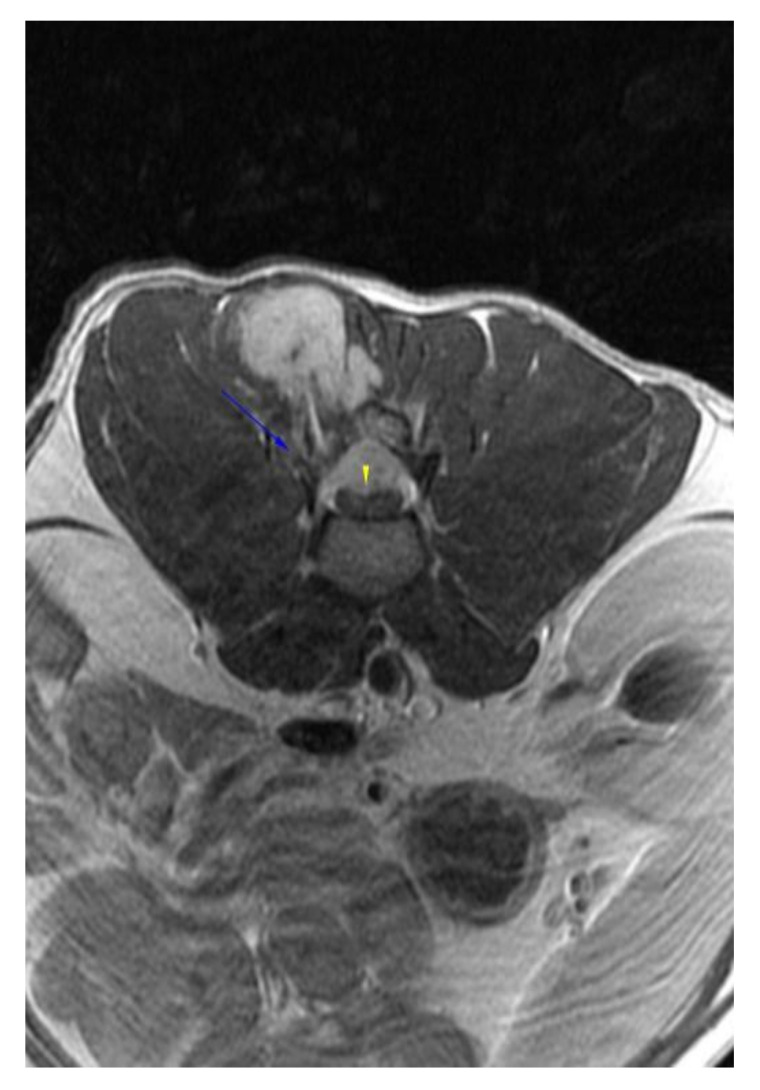
Post-contrast transverse T1W image showing the lysis of the right articular processes of L2/3 (blue arrow), expansion of the mass in the vertebral canal, and severe flattening of the spinal cord (yellow arrowhead).

**Figure 5 vetsci-09-00315-f005:**
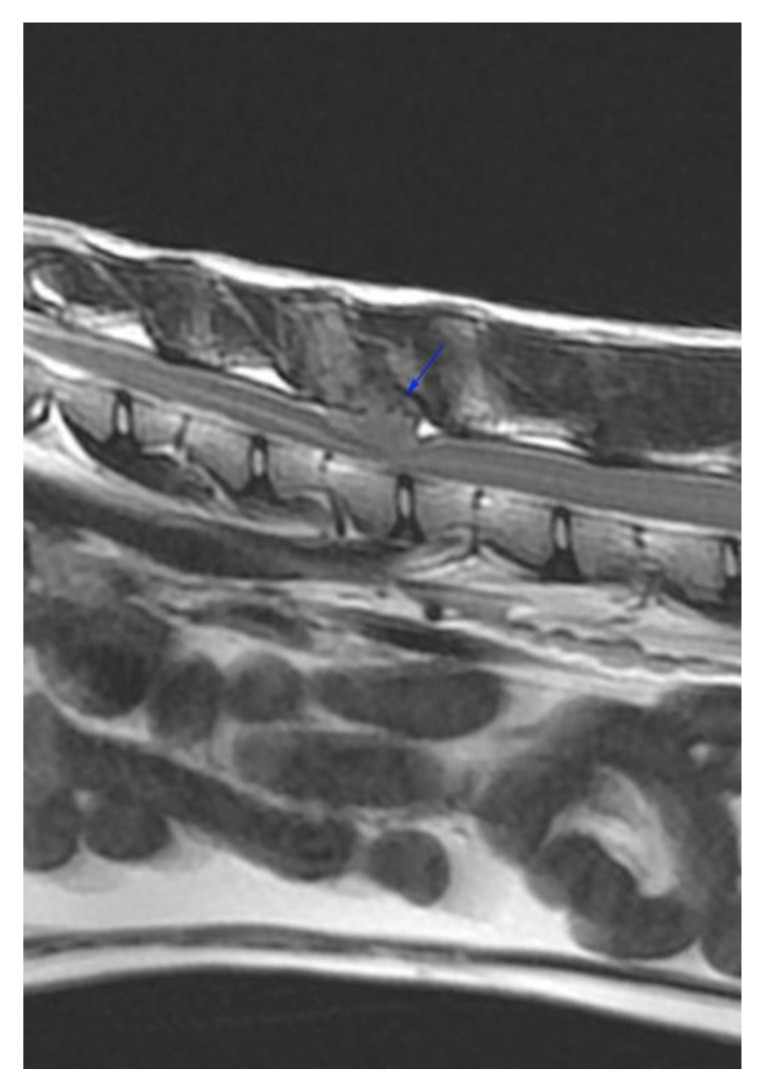
Sagittal T2W image of the spine showing the compression of the spinal cord by the extradural mass, which invades the dorsal lamina and spinous process of L2 (blue arrow). Note the dilation of the central canal and the hyperintensity of the spinal cord around the mass.

**Figure 6 vetsci-09-00315-f006:**
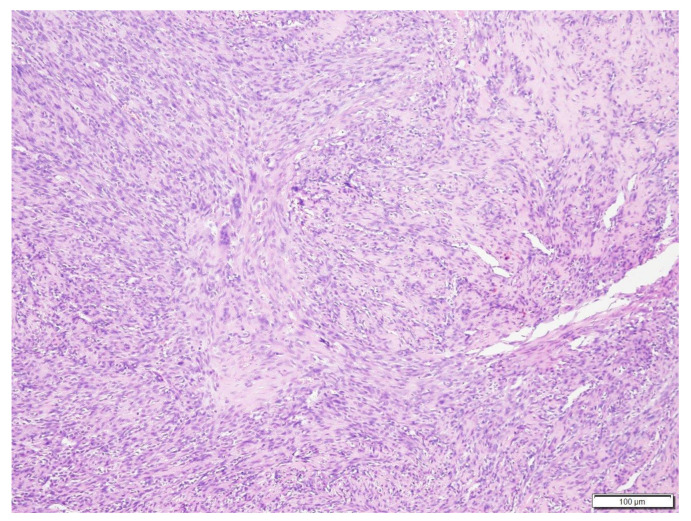
Histologic findings of the vertebral body mass. Neoplastic spindle cells are forming disorganized streams from fine to occasional dense fibrovascular stroma. Hematoxylin and eosin stain (×100 magnification).

**Figure 7 vetsci-09-00315-f007:**
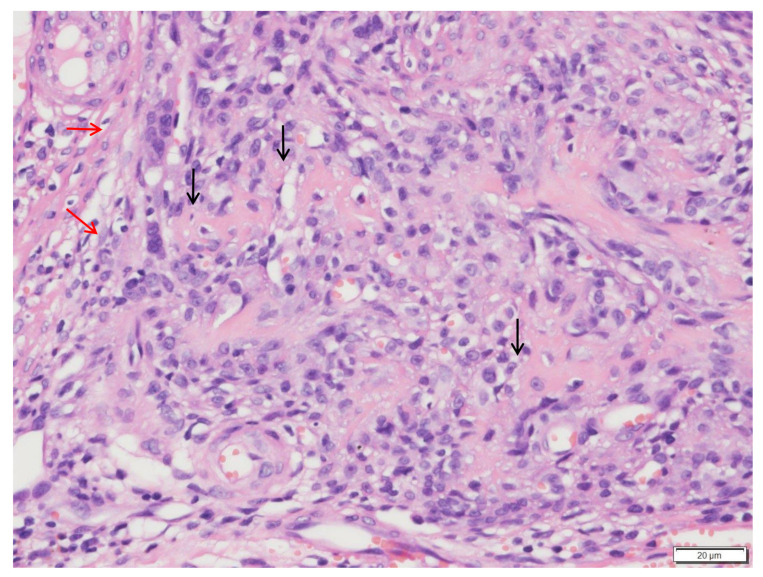
Histologic findings of the vertebral body mass. Occasional multinucleated cells (red arrows) and osteoid islands (black arrows) are noted. Hematoxylin and eosin stain (×400 magnification).

**Figure 8 vetsci-09-00315-f008:**
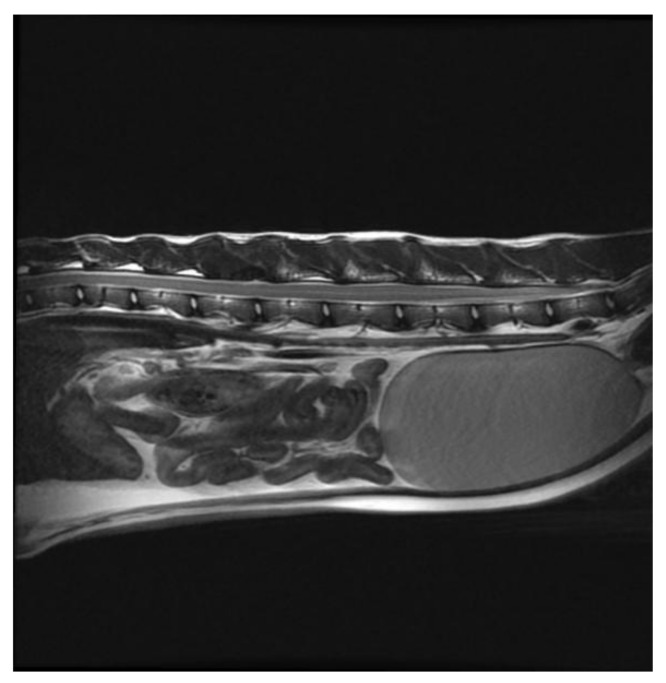
Sagittal T2W image of the spinal cord performed on 22 January 2022 showing no extradural and no epaxial mass.

## Data Availability

Not applicable.

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
