# Peer review of "Successful Treatment of Vertebral Osteosarcoma in a Cat Using Marginal Surgical Excision and Chemotherapy"

_vetsci, 2022, doi:10.3390/vetsci9070315_

Round 1

Reviewer 1 Report

An interesting case report quite simple and well documented. The authors describe a case of vertebral osteosarcoma in a cat and the follow up after surgery and adjuvant chemotherapy.

In the discussion the authors suggest a correlation between resolution of neoplastic lytic lesions with toceranib treatment.

The recover of the osteolytic lesion can also be related with alendronate treatment and not only with the use of toceranib, a tyrosine kinase inhibitor. The authors should add a paragraph related with the use of bisphosphonates, such as alendronate, a potent inhibitor of bone resorption, and the resolution of lytic lesions associated with bone metastasis.

The references should be reviewed according to the journal rules.

Author Response

Thank you for your comments. As you have suggested, we added this paragraph in the text and reviewed the reference.

"Bisphosphonates, such as alendronate, are a class of compounds used to treat osteoclast-mediated bone loss. These compounds are known to inhibit osteoclast activity but also to exhibit anti-angiogenic properties and anti-cancer effects in vitro. 

Bisphosphonates have multiple direct growth inhibitory effects on osteosarcoma cells through activation of apoptosis and inhibition of proliferation. Bisphosphonates have anti-angiogenic activity and direct inhibition on endothelial cells via inhibition of VEGF (vascular endothelial growth factor) and VEGFR [23]. However, significant clinical tumour response or survival improvement in both dogs and cats with osteosarcoma have never been reported.

In this patient, alendronate in combination with doxorubicin followed by toceranib treatment may have contributed to the resolution of the osteosarcoma bone lesions in this patient."

Under Conclusion

"Here, adjunctive alendronate, doxorubicin chemotherapy, followed by toceranib are likely to have contributed to the excellent outcome and the complete resolution of the residual tumour."

We will put the reference in as follows:

23) Ohba T, Cates JM, Cole HA, Slosky DA, Haro H, Ichikawa J, Ando T, Schwartz HS, Schoenecker JG. Pieotropic effects of bisphosphonates on osteosarcoma. Bone 2014 Jun, 63:110-20. DOI: 10.1016/j.bone.2014.03.005

Reviewer 2 Report

The manuscript by Giuliano et al entitled "Successful treatment of vertebral osteosarcoma in a cat using marginal surgical excision and chemotherapy" describes in detail a case of fibroblastic osteosarcoma with the involvement of muscles and nervous system. In this reviewer's opinion, the manuscript is well written and reports an interesting clinical case. However, before the publication of the manuscript, the authors need to add a histopathologic image of the tumor and a rational explanation for using oral toceranib phosphate. 

Author Response

Thank you for your comments. We have added a 100x and a 400x magnification of the vertebral osteosarcoma.

Regarding rational use of toceranib phosphate. We added this paragraph in the text:

Targeting VEGFR (vascular endothelial growth factor receptor) could decrease blood supply to the tumour, potentially reducing tumour growth. Toceranib also inhibits KIT (stem cell factor) and PDGFR (platelet derived growth factor) which have direct cytotoxic effects and contribute to the anti-angiogenic effect.

Reviewer 3 Report

The manuscript reported a case where surgery followed by chemotherapy completely removed a vertebral osteosarcoma without recurrence 16 months after the surgery in a 3-year-old cat. The manuscript is well-written. Below are my comments/questions

  1. Highlight in the abstract and/or introduction the significance of reporting this case.

2. Line 129-130: do you mean young dogs have worse prognosis?

3. Line 148: Typo. Should be “Its effect, if any, in feline osteosarcoma is not known”.

Author Response

Thank you for your comments.

In the abstract, we have added an additional sentence remarking the significance of reporting the case.

This is the first case report documenting the complete resolution of vertebral osteosarcoma neoplastic bone lesions after treatment with doxorubicin followed by toceranib phosphate. The treatment also resulted in the lack of tumour recurrence and an exceptionally long-term survival time.

Line 129-130: We have changed this to make it clearer as you have suggested.

In dogs with appendicular osteosarcoma, there is no clear difference in survival between young and old patients suffering from, but interestingly, young dogs could have a worse prognosis.

Line 148 Typo: We have changed this. I apologize for the missed typing error.

Its effect, if any, in feline osteosarcoma is not known.